# Is Influenza Vaccination Our Best ‘Shot’ at Preventing MACE? Review of Current Evidence, Underlying Mechanisms, and Future Directions

**DOI:** 10.3390/vaccines13050522

**Published:** 2025-05-14

**Authors:** Alexia El Khoury, Joy Abou Farah, Elie Saade

**Affiliations:** Division of Infectious Diseases & HIV Medicine, University Hospitals Cleveland Medical Center, School of Medicine, Case Western Reserve University, Cleveland, OH 44106, USA; joy.aboufarah@uhhospitals.org (J.A.F.); elie.saade@uhhospitals.org (E.S.)

**Keywords:** influenza vaccination, major adverse cardiovascular events (MACE), cardiovascular disease prevention, stroke prevention

## Abstract

Background: Major adverse cardiovascular events (MACE), including myocardial infarction, stroke, and cardiovascular death, are the leading contributors to global morbidity and mortality worldwide. Accumulating evidence suggests a strong association between influenza infection and increased risk of MACE, especially in high-risk populations. Influenza vaccination has been proposed as a potential strategy for reducing this risk by mitigating systemic inflammation and preventing atherosclerotic plaque destabilization, although the precise mechanisms remain under investigation. Results: Multiple meta-analyses and RCTs suggest that influenza vaccination is associated with a reduced risk of MACE, particularly in high-risk individuals with preexisting cardiovascular disease, but the results are less consistent for primary prevention in low-risk populations. The current data support the importance of early and annual vaccination for optimizing cardiovascular outcomes. Conclusions: Influenza vaccination is emerging as an effective and accessible strategy to reduce the risk of major adverse cardiovascular events, particularly in high-risk individuals. While further research is needed to clarify its role in low-risk populations and the underlying mechanisms of protection, current evidence supports its integration into cardiovascular preventive care.

## 1. Introduction

Major adverse cardiovascular events (MACE) are leading causes of death globally, accounting for millions of deaths each year, and placing a substantial burden on the healthcare system worldwide. MACE represent a cluster of critical clinical outcomes including acute myocardial infarction (AMI), stroke, and cardiovascular death, which constitute a crucial endpoint in cardiovascular research. Certain expanded definitions also include unstable angina, heart failure, and the need for revascularization [1]. These events are of significant concern given their association with increased morbidity and mortality and their substantial burden on healthcare costs; therefore, the identification of preventive strategies to reduce the incidence of MACE in at-risk individuals is of major importance.

Strong evidence suggests a significant relationship between influenza infection and MACE. Multiple studies have demonstrated that individuals infected with influenza are at increased risk of MACE [2]. Although the precise mechanisms remain incompletely understood, this risk is thought to be driven by the systemic inflammatory response and prothrombotic state induced by the influenza virus, which may exacerbate preexisting cardiovascular conditions or trigger new cardiovascular events in susceptible individuals [3]. Recapping these mechanisms is essential to understanding how influenza may act as a trigger for cardiovascular events.

Given this association, there is a growing interest in understanding the potential benefits of influenza vaccination as a preventive measure against MACE. Several observational studies and clinical trials have investigated the role of influenza vaccination in reducing cardiovascular events, particularly in high-risk populations, such as those with known cardiovascular disease (CVD) or heart failure [4,5]. The rationale is that by preventing influenza infection, vaccination may also reduce the risk of MACE, providing dual benefits for individuals at risk of both influenza and CVD. While current clinical guidelines recommend annual influenza vaccination for patients with cardiovascular conditions [6], the extent to which vaccination effectively reduces MACE remains an area of active research.

The aim of this narrative review is to synthesize the current evidence on the relationship between influenza vaccination and the prevention of MACE, summarize the possible mechanisms underlying this association, and discuss the implications for public health and clinical practice. This will be accomplished through the integration of findings from epidemiological studies, randomized controlled trials (RCTs), and meta-analyses, with the aim of providing a comprehensive overview of the role of influenza vaccination as a potential preventive strategy against MACE. Additionally, this review aims to highlight key research gaps and future directions for investigating the impact of vaccination on cardiovascular outcomes.

## 2. Mechanisms Linking Influenza Infection and MACE

Influenza infection has been consistently associated with an increased risk of MACE, such as AMI, heart failure exacerbations, and stroke. Specifically, research has identified a temporal correlation between influenza season and the occurrence of MACE, with studies consistently demonstrating an increased incidence of events such as AMI and stroke during the winter months, coinciding with peak influenza activity [7,8]. This observed seasonal pattern suggests a potential association between influenza infections and the precipitation of cardiovascular events. Notably, research conducted by Kwong et al. (2018) demonstrated that the risk of AMI was significantly elevated in the first week following laboratory-confirmed influenza infection, supporting the temporal association between these conditions [3]. Moreover, numerous meta-analyses, randomized controlled trials, and case–control studies indicate a decreased risk of MACE among patients who receive the influenza vaccine compared to those who do not [9,10,11].

The pathophysiological mechanisms contributing to this association are multifaceted and complex (Figure 1). First, influenza triggers a systemic inflammatory response, which includes the elevation of cytokines and acute-phase reactants, potentially destabilizing atherosclerotic plaques and promoting plaque rupture. This inflammatory state also impairs endothelial function, favoring vasoconstriction and thrombogenesis, thereby increasing the likelihood of acute coronary syndromes (ACSs).

Second, influenza infection induces a pro-thrombotic state by activating platelets and coagulation pathways, thereby increasing the risk of thrombotic events.

Furthermore, the infection induces substantial hemodynamic stress due to pyrexia, tachycardia, and hypoxemia, which may exacerbate myocardial oxygen demand, while simultaneously compromising oxygen supply, an imbalance that can precipitate ischemic events, particularly in individuals with underlying cardiovascular conditions.

In parallel, influenza may exert direct effects on cardiac tissues, potentially inducing myocarditis or pericarditis, which can impair cardiac function and precipitate arrhythmias or exacerbate heart failure [12].

Together, these overlapping inflammatory, thrombotic, hemodynamic, and direct viral mechanisms provide a strong biological rationale for the observed association between influenza infection and increase cardiovascular morbidity and mortality, highlighting the critical need for strategies that can mitigate these risks in vulnerable populations.

## 3. Mechanism of Action of Influenza Vaccine on MACE

Influenza virus has been shown to trigger MACE through multiple mechanisms. Primarily, preventing influenza infection by vaccination reduces systemic inflammation, a key factor in the destabilization and rupture of atherosclerotic plaques [13]. Additionally, since influenza infection induces a procoagulant state, increasing the risk of cardiovascular events, vaccination lowers the likelihood of clot formation, thereby reducing the risk of myocardial infarction (MI) and stroke [14,15]

Beyond infection prevention, several other mechanisms have been proposed to explain the cardioprotective effects of influenza vaccination (Figure 2). One hypothesis, suggested by Veljko et al., is that vaccine-induced antibodies act as agonists on atheroprotective pathways, particularly through the bradykinin 2 receptor (BKB2R), which plays a crucial role in cardiovascular homeostasis by functioning as an antiarrhythmic and antithrombotic agent. Furthermore, BKB2R activation enhances nitric oxide (NO) production, which regulates cardiac oxygen consumption, reduces oxidative stress, and mitigates endothelial dysfunction. The kallikrein–kinin system, acting through BKB2R, further provides antioxidant and anti-inflammatory protection against cardiovascular events [16].

In addition, Blanco-Colio LM et al. demonstrated that influenza vaccine antibodies may also modulate inflammation by reducing levels of the tumor necrosis factor-alpha-related weak inducer of apoptosis (TWEAK), a proatherogenic cytokine that stimulates smooth muscle cell proliferation and increases the synthesis of metalloproteinases, which in turn degrade plaque stability. By lowering TWEAK levels, influenza vaccination contributes to plaque stabilization, thus reducing cardiovascular risk [17].

Another proposed mechanism involves an autoimmune cross-reaction, as studies have identified a correlation between antibodies against influenza hemagglutinin A and oxidized LDL-lipoproteins in patients with rapidly progressing atherosclerosis, suggesting that the immune response to influenza may intersect with pathways involved in atherosclerosis progression [16,18].

Despite these findings, the specific cardiovascular effects of influenza vaccination remain controversial, and while promising results have been reported, further research is needed to clarify these connections and establish clear clinical pathways for the use of influenza vaccination in CVD prevention. Nevertheless, the demonstrated immunomodulatory effects of influenza vaccination highlight its potential as a promising strategy in the emerging field of “atherosclerosis vaccination.”

## 4. Effects of Influenza Vaccine on MACE

### 4.1. Effect of Vaccine in Specific Populations

#### 4.1.1. Effects of Influenza Vaccination in High-Risk Patients/Secondary Prevention of MACE

These proposed biological pathways provide a foundation for evaluating the real-world impact of influenza vaccination on MACE, especially in high-risk populations. Multiple studies have explored the impact of influenza vaccination on patients with CVD, yielding mixed results, as seen in Table 1.

##### Effect of Influenza Vaccination on Reducing AMI and Cardiovascular Death

The Flu Vaccination Acute Coronary Syndrome (FLUVACS) study revealed that vaccinated individuals had a notably lower risk of cardiovascular death compared to those who received a placebo. Similar reductions were observed in the composite triple endpoint of cardiovascular death, AMI, or severe recurrent ischemia (22% vs. 37%; HR 0.59, 95% CI: 0.4–0.86, *p* = 0.004). The most pronounced benefit was seen in patients with AMI, where cardiovascular mortality was 4% in the vaccine group vs. 21% in controls (HR 0.19, 95% CI: 0.07–0.53, *p* = 0.0002). No significant benefit was seen in the elective PCI subgroup [19].

Similarly, in a prospective randomized trial by Phrommintikul et al., at 12-month follow-up, the vaccine group experienced a significant reduction in major cardiovascular events, compared to controls (9.5% vs. 19.3%; adjusted HR: 0.67; 95% CI: 0.51–0.86; *p* = 0.005). Hospitalizations for ACS were also significantly reduced (4.5% vs. 10.6%; adjusted HR: 0.68; 95% CI: 0.47–0.98; *p* = 0.039). Although the incidence of cardiovascular death was lower in the vaccinated group (2.3% vs. 5.5%), this difference was not statistically significant (HR: 0.39; 95% CI: 0.14–1.12; *p* = 0.088) [20].

Conversely, the Influenza Vaccination in Secondary Prevention from Coronary Ischemic Events in Coronary Artery Disease (FLUCAD) did not show a statistically significant difference in MACE alone (3.0% vs. 5.87%; HR: 0.54; 95% CI: 0.24–1.21; *p* = 0.13). However, multivariable analysis identified influenza vaccination as an independent protective factor against coronary ischemic events (HR: 0.38; 95% CI: 0.19–0.78; *p* = 0.009), while cardiovascular mortality remained similar between groups (0.63% in vaccinated vs. 0.76% in placebo). The Efficacy of Influenza Vaccine in Reducing Cardiovascular Events in Patients with Coronary Artery Disease (IVCAD) trial also failed to show a decline in cardiovascular death or heart attacks after vaccination [21].

Moreover, the IAMI trial showed that administering the influenza vaccine within 72 h of a coronary procedure or AMI reduced MACE by 28%, cardiovascular mortality by 41%, and all-cause mortality by 41%, confirming its safety during the acute phase post-MI [9].

Beyond coronary events, some studies suggest a similar, although somewhat weaker association between influenza vaccination and reduced risks of heart failure, stroke, and transient ischemic attacks in the general population [27].

Given the considerable variability in the results of randomized and observational studies regarding the efficacy of the influenza vaccine for CVD prevention, on the efficacy of influenza vaccination for CVD prevention, the following meta-analyses have been conducted:

A Cochrane systematic review found a reduction in cardiovascular mortality among vaccinated patients with CVD. Although composite outcomes like MACE or coronary ischemic events were also reduced, the effects on individual outcomes such as AMI were not consistently significant. In contrast, data from the general population did not show statistically significant differences, likely due to small event numbers and underpowered studies [23]. Overall, the evidence supports the use of influenza vaccination for secondary prevention in patients with cardiovascular disease, while its role in primary prevention remains less clear.

Another meta-analysis highlighted a significant reduction in both cardiovascular deaths and MACE, although the reduction in AMI was not statistically significant (RR, 0.73; 95% CI, 0.49–1.09, *p* = 0.12) [22].

An updated meta-analysis of eight trials (14,420 patients) confirmed that influenza vaccination was associated with a significant 25% reduction in the risk of MACE, although differences in MI, CV death, and all-cause death individually did not reach statistical significance [2,24]. These findings reinforce influenza vaccination as a safe, effective, and underutilized strategy for secondary prevention in cardiovascular disease and post-MI care.

Interestingly, the impact of the influenza vaccine on reducing MACE and cardiovascular mortality in vaccinated individuals is comparable to, or in some cases exceeds, the efficacy of established cardiovascular therapies [28], such as statin, smoking cessation, beta blockers, and blood-eluting agents, as summarized in Table 2. The magnitude of benefit observed underscores the need to reposition influenza vaccination as not merely a preventive measure against respiratory infection but as a cardioprotective strategy equivalent in importance to pharmacologic therapies in both primary and secondary prevention settings, particularly in high-risk individuals with a recent history of MI.

##### Effect of Influenza Vaccination in Reducing Stroke

Furthermore, data from observational studies point to a possible protective effect of vaccination on stroke risk. A pooled analysis of 26 studies showed that influenza vaccination was associated with a 19% reduction in stroke incidence and hospitalization (OR: 0.81, 95% CI: 0.77–0.86, *p* = 0.00001). Among stroke patients, vaccinated individuals had a 50% lower risk of mortality (OR: 0.50, 95% CI: 0.37–0.68, *p* = 0.00001). Subgroup analyses revealed significant protective effects across comorbid populations, including those with atrial fibrillation (OR: 0.68), diabetes (OR: 0.76), COPD (OR: 0.70), and hypertension (OR: 0.76) [25]. In his large population-based observational cohort study, Holodinsky et al. reported a 22.5% reduction in overall stroke risk (HR: 0.775; 95% CI: 0.757–0.793) within 6 months post-vaccination, with consistent effect across all stroke subtypes (ischaemic, intracerebral hemorrhage, subarachnoid hemorrhage, and TIA), age groups, and sexes, regardless of baseline risk factors. Importantly, the risk of recurrent stroke was also reduced by 25.6% (HR: 0.744; 95% CI: 0.687–0.805) among individuals with a previous stroke [26] (Table 1).

#### 4.1.2. Effects of Influenza Vaccination in Low-Risk Patients/Primary Prevention/General Population of MACE

Evidence supporting the use of influenza vaccination for the primary prevention of CVD in low-risk or general populations is limited. RCTs face ethical challenges, due to the well-established benefits of influenza vaccination in preventing respiratory complications, and are further constrained by a low incidence of MACE in low-risk populations which require extremely large sample sizes to achieve sufficient statistical power. Consequently, the available research remains inconclusive.

A meta-analysis by Udell et al. showed that the influenza vaccine is associated with a 45% reduction in risk of MACE (RR 0.45, 95% CI: 0.32–0.63) as a secondary prevention. However, no significant risk reduction was found in patients without recent ACS (RR 0.94, 95% CI: 0.55–1.61) [38].

Additionally, a matched case–control study demonstrated that influenza vaccination is linked to a 19% reduced rate of first AMI [39].

In a large population-based self-controlled case series of 193,900 individuals aged 40–84 years in England, Davidson et al. (2023) found that influenza vaccination significantly reduced the risk of first acute cardiovascular events. The incidence ratio (IR) for cardiovascular events was 0.72 (95% CI: 0.70–0.74) during the 15–28 days post-vaccination, with protection persisting up to 120 days [IR 0.84 (95% CI: 0.82–0.85)]. The effect was consistent across age and cardiovascular risk strata, but more pronounced in younger adults (40–64 years: IR 0.54) and those with low baseline cardiovascular risk (QRISK2 < 10%: IR 0.48). Among event types, the greatest risk reduction was observed for myocardial infarction [IR 0.60 (95% CI: 0.57–0.64)]. These findings support the potential primary prevention benefit of influenza vaccination across a broad population, regardless of baseline cardiovascular risk [27].

In summary, the current data are insufficient to establish the definitive role of influenza vaccination in the primary prevention of MACE in the general population. Further research is needed to clarify its potential benefits in low-risk individuals.

### 4.2. Timing of Influenza Vaccination and Cardiovascular Events

#### 4.2.1. Optimal Seasonal Timing of Vaccination

The seasonality of influenza varies across countries and must be better characterized to optimize the timing of vaccination [40]. The optimal timing of influenza vaccination depends on several factors: the rate at which vaccine effectiveness wanes, the initial effectiveness of the vaccine, and the timing of the seasonal influenza peak. An analysis by Spencer et al. found that delaying vaccination can be beneficial across all age groups—particularly in seasons with a later peak or when vaccine effectiveness declines rapidly [41]. The protective effect of influenza vaccination against MACE may reflect this relationship, as it is likely mediated through the prevention of influenza infection. However, we found only one study that directly examined the timing of influenza vaccination in relation to AMI risk. In the United Kingdom, early seasonal influenza vaccination, administered before mid-November, is associated with a more pronounced reduction in AMI risk (adjusted OR 0.79; 95% CI: 0.75–0.83) compared to later vaccination given after mid-November [39]. Given that influenza seasonality varies by region, these findings warrant validation in other settings with different seasonal patterns.

#### 4.2.2. Post-Cardiovascular Events or Hospitalization Timing

While optimizing the timing of seasonal vaccination is critical at the population level, another key opportunity for influenza prevention lies in targeting high-risk individuals during periods of increased vulnerability. Hospitalization, particularly for cardiovascular conditions, represents a crucial window to administer influenza vaccination and potentially reduce subsequent MACE. This approach complements seasonal strategies by ensuring protection in patients who may have missed early vaccination or who remain at elevated risk despite community-level efforts.

The IAMI trial investigated the effect of early influenza vaccination administered during hospitalization (within 72 h of coronary angiography or admission for myocardial infarction). It demonstrated that early vaccination after MI reduced the risk of MACE by 28%, cardiovascular mortality by 41%, and all-cause mortality by 41% in patients with acute or high-risk coronary artery disease, significantly improving outcomes at 12 months compared to the placebo. This effect on all-cause death at one year was more pronounced in the group receiving early vaccination (HR 0.50; 95% CI, 0.29 to 0.86) compared to the late vaccination group (HR 0.75; 35% CI, 0.40 to 1.40). While the positive trend for reducing MI risk was not statistically significant, likely due to low event numbers, the study confirmed the safety of influenza vaccination during the acute phase after MI, supporting its use in all CVD patients regardless of timing [9].

Despite strong guideline recommendations, influenza vaccination remains underused in this population [42]. In-hospital administration not only ensures timely immunization but also enhances patient adherence [43], and may deliver cardiovascular protection comparable to other standard therapies, such as statins or ACE inhibitors, as shown earlier. These findings support the integration of influenza vaccination into standard post-MI care pathways, particularly during flu season.

#### 4.2.3. Impact of Annual Vaccination on MACE

While early vaccination following hospitalization offers timely protection during a particularly vulnerable period, long-term cardiovascular benefits may also be achieved through consistent annual influenza vaccination. Repeated yearly immunization not only maintains protective immunity against evolving influenza strains but may also contribute to sustained reductions in the risk of MACE over time, especially in individuals with established cardiovascular disease.

In a large Danish nationwide cohort study of over 134,000 patients with newly diagnosed heart failure, Modin et al. demonstrated that receiving influenza vaccination after diagnosis was associated with an 18% reduction in both all-cause and cardiovascular mortality after adjusting for comorbidities, medications, socioeconomic status, and other confounders. Notably, annual vaccination conferred even greater protection, with a 19% reduction in mortality, compared to those vaccinated less frequently. The study also showed a dose–response relationship, where a greater cumulative number of vaccinations was linked to progressively lower mortality risk (*p* for trend < 0.001). Additionally, vaccinations administered earlier in the flu season were associated with larger reductions in mortality compared to those given later [44]. Similarly, stroke risk in individuals who received the influenza vaccine repeatedly across multiple years was reduced compared to those who received a single vaccination. This pattern was consistent across all stroke subtypes and demographic subgroups, including different ages, sexes, and risk factor profiles [26]. This emphasizes the importance of consistent and timely annual influenza vaccination for optimal cardiovascular protection.

### 4.3. Type of Vaccine and Dosage Considerations

#### 4.3.1. Standard-Dose vs. High-Dose Influenza Vaccination

Several studies have investigated whether high-dose influenza vaccines offer superior cardiovascular protection compared to standard-dose formulations, particularly in patients with underlying cardiovascular disease.

The INVESTED trial, a large randomized controlled study of over 9000 patients with recent myocardial infarction or heart failure hospitalization, compared a high-dose trivalent influenza vaccine with a standard-dose quadrivalent vaccine. While the high-dose vaccine elicited stronger immune responses, it did not result in a statistically significant reduction in the composite outcome of all-cause mortality or cardiopulmonary hospitalizations compared to the standard-dose vaccine (HR 1.06; 95% CI: 0.97–1.17) [45].

Similarly, in a post hoc analysis of nursing home populations, Saade et al. reported that high-dose vaccination did not significantly reduce MACE compared to standard-dose vaccination (HR 0.88; 95% CI: 0.76–1.01), although a reduction in respiratory-related hospitalizations was observed in certain subgroups (HR = 0.87 (95% CI: 0.77–0.97)) [46].

Additionally, the VIP-ACS trial assessed two dosage strategies using the same vaccine (Fluarix^®^): a standard-dose quadrivalent inactivated influenza vaccine versus a double-dose vaccine administered at different time points. The trial reported no significant differences in all-cause mortality, MACE, or cardiopulmonary outcomes between groups, and the safety profiles were similar [47].

Collectively, these findings suggest that increasing the vaccine dose does not consistently confer additional cardiovascular protection in high-risk populations.

#### 4.3.2. Adjuvanted Vaccines and Other Considerations

Additional research on vaccine formulations includes a study conducted in Italy, where the adjuvanted influenza vaccine MF59-TIV was found to reduce the risk of hospitalizations for pneumonia and cerebrovascular/cardiovascular events compared to non-adjuvanted trivalent vaccines [48].

Overall, while influenza vaccination is strongly recommended for patients with CVD, current evidence does not indicate a clear preference for any specific vaccine type or dosage. Therefore, any approved influenza vaccine formulation is suitable for these patients, with the emphasis remaining on timely and effective immunization.

## 5. Future Directions and Research Gaps

While evidence increasingly supports the beneficial role of influenza vaccination in reducing the risk of MACE, several key research gaps remain.

Foremost among these is the need for large-scale, adequately powered RCTs to specifically evaluate the impact of influenza vaccination on cardiovascular outcomes across diverse patient populations, including those without established CVD but at risk for MACE. Additionally, the role of vaccination in stroke prevention requires further exploration; although observational studies indicate potential benefits, no large RCTs have directly targeted patients with recent cerebrovascular events. The existing studies vary widely in the populations studied, definitions of cardiovascular events, and timing of vaccination, creating heterogeneity that complicates the ability to draw definitive conclusions.

Another area needing investigation is the precise mechanisms through which influenza vaccination influences cardiovascular outcomes.

Timing is also another crucial factor; while trials like IAMI suggest that early vaccination (within 72 h post-MI or coronary procedure) may be advantageous, more robust studies are required to confirm the optimal timing for maximizing cardiovascular protection. Furthermore, the long-term effects of repeated annual influenza vaccinations on cardiovascular health remain unclear, underscoring the need for research on cumulative impact and patient outcomes over time.

Another gap lies in exploring different delivery methods for the vaccine to optimize cardiovascular benefits, which also represents an untapped research avenue. Additionally, understanding the impact of vaccine type and dosage is essential; the INVESTED trial showed no added benefit of high-dose vaccines over standard-dose vaccines, and the efficacy of adjuvanted vaccines like MF59-TIV in specific subgroups remains undetermined.

Addressing these research gaps is crucial for developing more targeted vaccination strategies and enhancing the understanding of how influenza vaccination can be effectively integrated into CVD prevention. Meanwhile, healthcare providers and policymakers should consider the growing evidence and prioritize influenza vaccination for patients with recent CVDs as a feasible and potentially life-saving preventive measure.

## 6. Conclusions

Influenza vaccination may indeed be one of our best “shots” at preventing MACE. The growing body of evidence suggests that timely and annual vaccination can significantly reduce the risk of cardiovascular events, particularly in high-risk patients. While questions remain about its impact on low-risk populations, the optimal timing, and the mechanisms involved, the protective benefits are clear, potentially rivaling established cardiovascular therapies. As research continues to close the gaps in our understanding, influenza vaccination stands out as a safe, effective, and widely accessible strategy for cardiovascular protection. For those with CVD, this simple intervention could be a crucial step toward better heart health and reduced risk of MACE.

## Figures and Tables

**Figure 1 vaccines-13-00522-f001:**
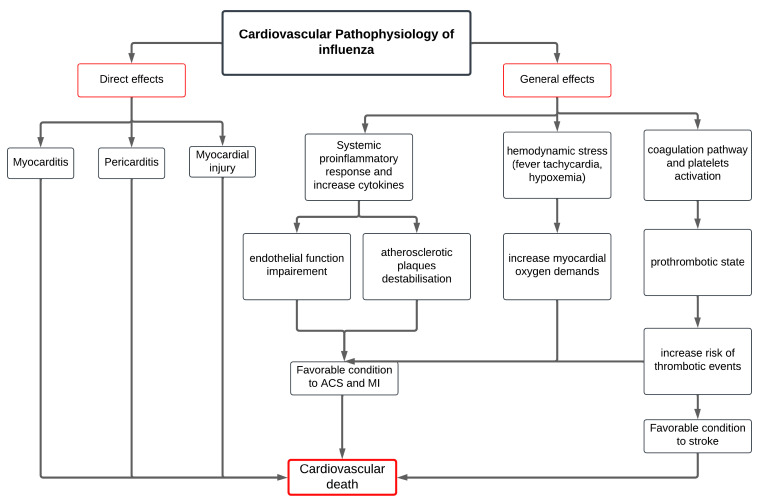
Proposed pathway describing the causal link between influenza infection and major adverse cardiovascular events.

**Figure 2 vaccines-13-00522-f002:**
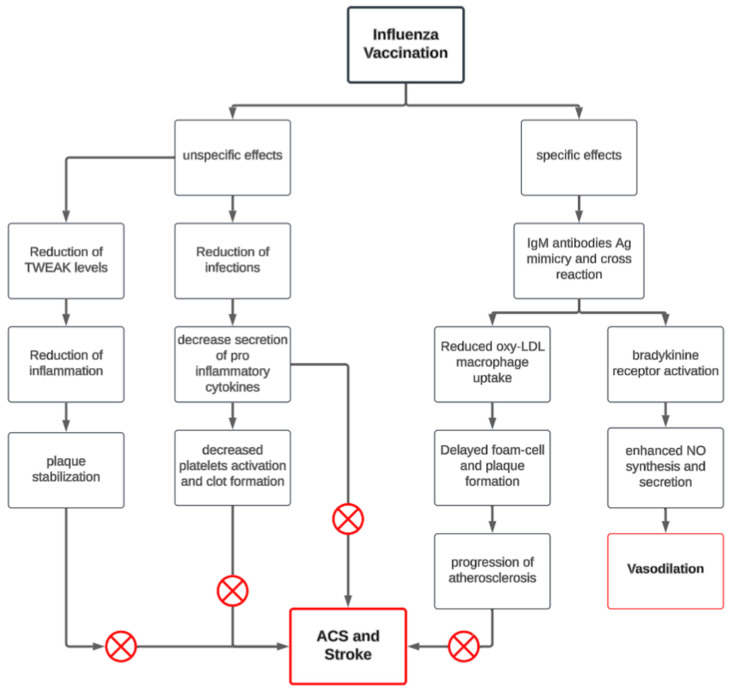
Mechanisms behind how the influenza vaccine protects against MACE.

**Table 1 vaccines-13-00522-t001:** Summary of cardiovascular and stroke outcomes in key studies evaluating the impact of influenza vaccination on secondary prevention of MACE.

Outcome	Study	Study Type	Results	Statistics
Cardiovascular death	FLUVACS [19]	RCT	Significant reduction	HR = 0.34 (95% CI: 0.17–0.71), *p* = 0.002
	Phrommintikul et al. [20]	RCT	No significant reduction	unadjusted HR 0.39 (95% CI: 0.14–1.12), *p* = 0.088
	IVCAD [21]	RCT	No significant reduction	(29% vs. 26%, *p* = 0.60)
	IAMI [9]	RCT	Significant reduction	HR = 0.59 (95% CI, 0.39–0.90), *p* = 0.014
	FLUCAD [4]	RCT	No significant reduction	HR = 1.06 (95% CI: 0.15–7.56), *p* = 0.95
	Yedlapati et al. [22]	Meta-Analysis	Significant reduction	RR = 0.82 (95% CI: 0.80–0.84), *p* < 0.001
	Clar et al. [23]	Meta-Analysis	Significant reduction	RR = 0.45 (95% CI: 0.26–0.76), *p* = 0.003
MACE	Phrommintikul et al. [20]	RCT	Significant reduction in ACS, HF, stroke	HR 0.70 (95% CI: 0.57–0.86), *p* = 0.088
	IAMI [9]	RCT	Significant reduction in AMI	HR = 0.72 (95% CI, 0.52–0.99), *p* = 0.040
	Clar et al. [23]	Meta-Analysis	No significant reduction in MI	
	Yedlapati et al. [22]	Meta-Analysis	Significant reduction	RR = 0.87 (95% CI: 0.80–0.94), *p* < 0.001
	Maniar et al. [24]	RCT	Significant reduction in MACE risk but No significant reduction for MI	RR 0.75 (95% CI: 0.57–0.97), *I*^2^ = 56% (RR, 0.73; 95% CI, 0.52–1.10, *I*^2^ = 0%)
	FLUVACS [19]	RCT	Significant reduction in MI	HR = 0.59 (95% CI 0.4—0.86), *p* = 0.004
	FLUCAD [4]	RCT	Significant reduction in coronary ischemic events	HR 0.54 (95% CI: 0.24–1.21), *p* = 0.13
	Zahhar et al. [25]	Systematic Review/Meta-Analysis	Significant reduction in stroke events	OR = 0.81, 95% CI [0.77–0.86], *p* = 0.00001
	Holodinsky et al. [26]	Self-controlled case series	Significant reduction in stroke events	HR = 0.775 [95% CI 0.757–0.793]

**Table 2 vaccines-13-00522-t002:** Efficacy of conventional coronary prevention interventions and influenza vaccine in the prevention of major adverse cardiovascular events and/or mortality.

Intervention	Cardiovascular Death	Primary Prevention of MI	Secondary Prevention of MI	Prevention of Stroke
Smoking cessation	14.8–40% [29,30]	11% [30]	14.8–36% [29,31]	30% [32]
Statin	4–8% [33,34]	21–27% [35]	13–27% [33,34]	12–17% [33,34]
Beta blockers	15–31% [3,36]	-	9% [36]	-
Blood-eluting agents	12% [37]	-	19% [37]	39% [37]
Influenza vaccine	18–55%	6–19%	13–46%	19–25%

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
