# Peer review of "Is Influenza Vaccination Our Best ‘Shot’ at Preventing MACE? Review of Current Evidence, Underlying Mechanisms, and Future Directions"

_vaccines, 2025, doi:10.3390/vaccines13050522_

Round 1

Reviewer 1 Report

Comments and Suggestions for Authors

This is a review on influenza vaccination and MACE. Here, not only the efficacy of the vaccine on CV events is reviewed, but also possible mechanisms of action.

There are some issues the authors should revise:

Title and introduction: As there are already several reviews on the effects of influenza vaccines on MACE, you could emphasize here that you also recap the possible mechanisms of action.

Abstract: What is written in the conclusion section rather belongs to the background or methods. Please find another conclusion for the abstract.

In section 3, the mechanisms are described. They are also shown in fig. 2, except one (TWEAK). You could include this in the figure, too.

In 4.2.1, you mention the timing of vaccination, before or after mid-November.  This is taken from the cited paper, but cannot be transfered to influenza vaccination as such. As we know, influenza sesonality varies from region to region, and the data from the cited publication are true for UK. So you should abstract this for a general statement.

Abbreviations: Please bring them into an alphabetical order.

Author Response

Thank you very much for taking the time to review this manuscript. Please find detailed responses below and the corresponding revisions/corrections in track changes in the re-submitted files.

Comments 1: Title and introduction: As there are already several reviews on the effects of influenza vaccines on MACE, you could emphasize here that you also recap the possible mechanisms of action.

Response 1: Thank you for pointing this out. We agree with this comment. Therefore, we have revised the title and introduction to better emphasize that, in addition to reviewing the clinical evidence on influenza vaccination and MACE, we also explore the proposed mechanisms of action in detail. [Is Influenza Vaccination Our Best 'Shot' at Preventing MACE? Review of Current Evidence, Underlying mechanisms and Future directions.] (page 1 line 2)

Comments 2: Abstract: What is written in the conclusion section rather belongs to the background or methods. Please find another conclusion for the abstract.

Response 2: Agree. We have, accordingly, rephrased the conclusion to more appropriately summarize the key takeaways. [Influenza vaccination is emerging as an, effective, and accessible strategy to reduce the risk of major adverse cardiovascular events, particularly in high-risk individuals. While further research is needed to clarify its role in low-risk populations and the underlying mechanisms of protection, current evidence supports its integration into cardiovascular preventive care.] (page 1 lines 27-31)

Comments 3: In section 3, the mechanisms are described. They are also shown in fig. 2, except one (TWEAK). You could include this in the figure, too.

Response 3: Thank you for pointing this out. We have updated Figure 2 to include the TWEAK pathway, ensuring alignment with the mechanisms discussed in the text. (p6 line 151)

Comments 4: In 4.2.1, you mention the timing of vaccination, before or after mid-November.  This is taken from the cited paper, but cannot be transfered to influenza vaccination as such. As we know, influenza sesonality varies from region to region, and the data from the cited publication are true for UK. So you should abstract this for a general statement.

Response 4: Agree. We have, accordingly modified the wording to reflect the regional variability in influenza seasonality and clarified that the cited data apply specifically to the UK, while generalizing the message for broader relevance. [The seasonality of influenza varies across countries and must be better characterized to optimize the timing of vaccination [40]. The optimal timing of influenza vaccination depends on several factors: the rate at which vaccine effectiveness wanes, the initial effectiveness of the vaccine, and the timing of the seasonal influenza peak. An analysis by Spencer et al found that delaying vaccination can be beneficial across all age groups—particularly in seasons with a later peak or when vaccine effectiveness declines rapidly [41]. The protective effect of influenza vaccination against MACE may reflect this relationship, as it is likely mediated through the prevention of influenza infection. However, we found only one study that directly examined the timing of influenza vaccination in relation to AMI risk.  In the United Kingdom, early seasonal influenza vaccination, administered before mid-November, is associated with a more pronounced reduction in AMI risk (adjusted OR 0.79; 95% CI: 0.75–0.83) compared to later vaccination given after mid-November. Given that influenza seasonality varies by region, these findings warrant validation in other settings with different seasonal patterns.] (p 10 lines 297-312)

Comments 5: Abbreviations: Please bring them into an alphabetical order.

Response 5: All abbreviations have been reordered alphabetically as requested (p.13- 14)

Reviewer 2 Report

Comments and Suggestions for Authors

Comments: Major revision required 

  1. General comments and suggestions for improvement:

    Overall, the article provides relevant and valuable insights; however, the paragraph organization needs significant improvement to enhance readability and logical flow.

    2.  The review should more clearly elaborate the main reasons for the strong association between influenza infections and major adverse cardiovascular events (MACE). The current explanation lacks depth and coherence.

    3. The article should address in more detail why there is limited clinical trial data to support the efficacy of influenza vaccination for primary prevention of MACE in low-risk or general populations. A discussion of the ethical and logistical challenges of conducting such studies would add value and context.

    4. Figure 2 – Mechanistic insights: Figure 2, titled “Mechanisms Behind How the Influenza Vaccine Protects Against MACE,” needs further explanation. As the manuscript relates to COPD, it would be beneficial to include the cytokine network and inflammatory cascade in this section to explain the underlying mechanisms more fully for better understanding. 

    5. The impact of influenza vaccination on MACE during COVID-19 infection should be presented in a separate subsection. This would help to emphasize the importance of co-infections and the potential cardiovascular benefits of vaccination in such situations.

    6. The article should discuss the efficacy of influenza vaccination in immunocompromised patients with cardiac complications. This subgroup is of particular clinical importance and deserves special attention.

    7. The manuscript requires professional proofreading to correct numerous typographical and grammatical errors.

    8. In addition, all references should be revised to comply with MDPI formatting guidelines.

Comments on the Quality of English Language

 The manuscript requires professional proofreading to correct numerous typographical and grammatical errors.

Author Response

We sincerely thank the reviewer for their thoughtful and constructive comments. We appreciate the time and effort dedicated to reviewing our manuscript and for providing valuable suggestions that have helped us improve the quality and clarity of our work.

Comment 1: The paragraph organization needs significant improvement to enhance readability and logical flow.

Response 1: Agree. We have accordingly improved the organisation. We added linking paragraphs for better flow, and the large subsection has been divided into two distinct subsections to enhance clarity and readability. [4.1.1.1. Effect of influenza vaccination in reducing AMI and cardiovascular death – p. 7 line 169] [4.1.1.2. Effect of influenza vaccination in reducing Stroke – p. 9 line 243]

Comment 2: The review should more clearly elaborate the main reasons for the strong association between influenza infections and MACE. The current explanation lacks depth and coherence.

Response 2: We thank the reviewer for this important comment. We agree that further elaboration would strengthen the manuscript. Accordingly, we have developed and expanded the relevant paragraph to provide a more comprehensive and coherent explanation of the multifactorial mechanisms linking influenza infection to increased cardiovascular risk. (p.3, lines 93 to 111)

Comment 3:
The article should address in more detail why there is limited clinical trial data to support efficacy in low-risk/general populations.

Response 3: Thank you for pointing this out. We have expanded the discussion by emphasizing the ethical challenges and the low incidence of events in low-risk populations, which make clinical trials difficult to conduct. We have also added another relevant study to this paragraph. [RCTs face ethical challenges, due to well established benefits of influenza vaccination in preventing respiratory complications, and are further  constrained by a low incidence of MACE in low-risk populations which requires extremely large sample sizes to achieve sufficient statistical power – p. 10 lines 270-274] – [In a large population-based self-controlled case series of 193,900 individuals aged 40–84 years in England, Davidson et al. (2023) found that influenza vaccination signifi-cantly reduced the risk of first acute cardiovascular events. The incidence ratio (IR) for cardiovascular events was 0.72 (95% CI: 0.70–0.74) during the 15–28 days post-vaccination, with protection persisting up to 120 days [IR 0.84 (95% CI: 0.82–0.85)]. The effect was consistent across age and cardiovascular risk strata, but more pronounced in younger adults (40–64 years: IR 0.54) and those with low baseline cardiovascular risk (QRISK2 <10%: IR 0.48). Among event types, the greatest risk reduction was observed for myocardial infarction [IR 0.60 (95% CI: 0.57–0.64)]. These findings support the potential primary prevention benefit of influenza vaccination across a broad population, regardless of baseline cardiovascular risk – p. 10 lines 281-291]

Comment 4: Figure 2 needs further explanation. As the manuscript relates to COPD, it would be beneficial to include the cytokine network and inflammatory cascade.

Response 4: We respectfully clarify that the manuscript does not focus on COPD. The subject of our review is the impact of influenza vaccination on MACE broadly, not within a COPD-specific population. Therefore, we did not expand on COPD-specific mechanisms.

Comment 5: The impact of influenza vaccination on MACE during COVID-19 infection should be presented in a separate subsection.

Response 5: We respectfully note that COVID-19 and co-infections are beyond the scope of this review, which specifically focuses on the association between influenza vaccination and cardiovascular outcomes. For this reason, we have not created a separate subsection on COVID-19.

Comment 6: The article should discuss the efficacy of influenza vaccination in immunocompromised patients with cardiac complications.

Response 6: We thank the reviewer for this insightful comment. We agree that immunocompromised patients represent an important subgroup at increased risk for influenza infection and its complications. However, our primary objective was to review the effect of influenza vaccination on major adverse cardiovascular events (MACE) specifically, including individuals at both high and low cardiovascular risk. this being said, we did not identify studies that clearly evaluated the impact of influenza vaccination on MACE outcomes specifically in immunocompromised populations. We believe that this remains an important area for future research.

Comment 7: The manuscript requires professional proofreading to correct numerous typographical and grammatical errors.

Response 7: We thank the reviewer for highlighting this. The manuscript has been carefully proofread, and typographical and grammatical errors have been corrected.

Comment 8: All references should be revised to comply with MDPI formatting guidelines.

Response 8: All references have been revised and formatted according to MDPI guidelines.

Reviewer 3 Report

Comments and Suggestions for Authors

Thank you for allowing me to review this interesting association MACE and indirect protection by influenza vaccination. It is well known that influenza vaccines are not ideal because infection is many times left unprotected , but the fact that severe outcomes related to influenza can be prevented is of utmost importance to be highlighted in order to achieve greater acceptance of the vaccine despite low effectiveness in disease infection prevention. Being MACDE one of the leading causes of mortality mekes this work even more relevant.

Here are some considerations:

Title is correct , although I would add "Review" of current evidence......  for clarity

Abstract: line 16  change Meta-analyses to meta-analyses

Introduction: Why do the authors start out with Ischemic heart disease (IHD) , it should be included into the MACE category . Besides it is not mentioned anymore throughout the paper. Change the order and start the paragrapph with lines 33 and iclude IHD in definition or delete

Table headings should preceed tables

Author Response

Thank you very much for your thoughtful and supportive review. We greatly appreciate your recognition of the importance of this topic and your valuable suggestions to improve the clarity and quality of the manuscript.

Comment 1: Title is correct, although I would add "Review of current evidence" for clarity.

Response 1: Thank you for the suggestion. As recommended, we have revised the title to include “Review of current evidence” for greater clarity and accuracy. [Is Influenza Vaccination Our Best 'Shot' at Preventing MACE? Review of Current Evidence, Underlying mechanisms and Future directions.] (page 1 line 2)

Comment 2: Abstract: line 16 — change Meta-analyses to meta-analyses.

Response 2: We appreciate your careful review. We have corrected "Meta-analyses" to lowercase as suggested.  (p.1, line 18)

Comment 3: Introduction: Why do the authors start out with Ischemic heart disease (IHD)? It should be included in the MACE category. Besides, it is not mentioned anymore throughout the paper. Change the order and start the paragraph with lines 33 and include IHD in the definition or delete.

Response 3: Thank you for this observation. To maintain consistency and improve the logical flow, we have removed the reference to ischemic heart disease (IHD) from the introduction, as it was not further discussed throughout the manuscript. (p. 2 line 38-40)

Comment 4:

Table headings should precede tables.

Response 4: We have revised the formatting so that all table headings now precede their respective tables.

Round 2

Reviewer 1 Report

Comments and Suggestions for Authors

The paper has improved a lot.

Two further small issues:

line 333: The wording is confused, please revise this sentence.

figure 2: box "platelet activation and clot formation": it must be "decreased ..." or "no ..."

Author Response

Thank you once again for your thoughtful feedback.

  • The sentence on line 333 has been revised for clarity.
  • The label in Figure 2 under "platelet activation and clot formation" has also been corrected as mentionned "decreased platelet actiavtion".

We appreciate your careful review and hope the revised manuscript meets your expectations.

Reviewer 2 Report

Comments and Suggestions for Authors

Yes, it is much improved now; however, the references are still not formatted according to MDPI guidelines. 

Can be accepted

Author Response

Thank you for your feedback and for your positive assessment of the revised manuscript.

We have now carefully corrected the reference list. We hope it meets the required standards this time.

We truly appreciate your time and consideration.